# 3D printing of different fibres towards HA/PCL scaffolding induces macrophage polarization and promotes osteogenic differentiation of BMSCs

Jiaxiang Song[1☯], Shuai Huang[2☯], Xitao Linghu[1☯], Hengpeng Wu[1,3‡], Zhenyu Wen[4‡], Xiang Li[1‡], Qiping Huang[1‡], Weikang Xu[3,5]*, Qingde Wa[1]*

1 Department of Orthopaedic Surgery, The Second Affiliated Hospital of Zunyi Medical University, Zunyi, Honghuagang District, Guizhou, China, 2 Department of Orthopaedic Surgery, The Second Affiliated Hospital of Guangzhou Medical University, Guangzhou, Haizhu District, Guangdong, China, 3 Institute of biological and Medical Engineering, Guangdong Academy of Sciences, Haizhu District, Guangzhou, Guangdong, China, 4 Zunyi Medical University, Xinpu New District, Zunyi City, Guizhou, China, 5 Guangdong Institute of Medical Instruments, National Engineering Research Center for Healthcare Devices, Guangdong Provincial Key Laboratory of Medical Electronic Instruments and Materials, Tianhe District, Guangzhou, Guangdong, China

☯ These authors contributed equally to this work.
‡ These authors also contributed equally to this work
* 759200816@qq.com (WX); wqd887zsy@126.com (QW)

**Data Availability Statement:** All relevant data are within the manuscript and its Supporting Information files.

## Abstract

With the rise of bone tissue engineering (BET), 3D-printed HA/PCL scaffolds for bone defect repair have been extensively studied. However, little research has been conducted on the differences in osteogenic induction and regulation of macrophage (MPs) polarisation properties of HA/PCL scaffolds with different fibre orientations. Here, we applied 3D printing technology to prepare three sets of HA/PCL scaffolds with different fibre orientations (0–90, 0-90-135, and 0-90-45) to study the differences in physicochemical properties and to investigate the response effects of MPs and bone marrow mesenchymal stem cells (BMSCs) on scaffolds with different fibre orientations. The results showed that multi-angle staggered fibres affected the overall porosity and compressive strength of the scaffolds. Compared with the other two groups, the 0-90-45 scaffold induced osteogenic differentiation of BMSCs more significantly, while promoting the polarisation of MPs towards the M2 phenotype to form an osteogenic-friendly immune microenvironment. Unexpectedly, the 0-90-45 scaffold significantly upregulated the expression of angiogenic genes (PDGF, VEGF). Therefore, we conclude that the multi-angle interlaced fibres better mimic the physiological structure of cancellous bone, and that the excellent biomimetic properties reflect the best in vitro osteogenic, immunomodulatory and angiogenic effects. In conclusion, this study is a step forward in the exploration of BET scaffolds and provides a very promising bone filling material.

**Funding:** This research was supported by the Key Program for Science and Technology Project of Guizhou Province (ZK [2021] 007), the Program for Science and Technology Project of Guizhou Province, Qiankehe Platform Talents ([2021] 5613), the Guangdong Province Science and Technology Plan Project (2024A1515012265 and 2022A1515140193), the Guangzhou Science and Technology Plan Project (2023A03J0405).

**Competing interests:** The authors have declared that no competing interests exist.

## Introduction

Regeneration of critical-size bone defects caused by infection, fracture, congenital disease and tumor resection remains a major clinical challenge, often relying on bone grafting [1]. With the rise of bone tissue engineering (BET), 3D printing is widely used in the study of bone repair materials by simulating the bionic structure of bone with the aid of imaging data and computer-aided design models to directly fabricate highly accurate three-dimensional solids layer by layer [2–4].

Polycaprolactone (PCL) is approved for clinical use by the U.S. Food and Drug Administration (FDA) and is often used as a raw material for 3D printing because of its excellent biocompatibility, plasticity, and slow degradation compared to other materials [5,6]. However, PCL itself lacks biological sites, and application to bone repair often requires modified or composite bioceramics such as hydroxyapatite (HA) [7,8]. HA is the main inorganic component of human bone, which is implanted in the body to release more calcium and phosphorus components to help bone mineralisation. During the process of vascularization, these biodegradation products can act as signaling molecules to stimulate the proliferation and migration of vascular endothelial cells, which in turn promotes the formation of new blood vessels [9]. For example, Jiayi Ma et al. in studying the effects of HA on human umbilical cord vein endothelial cells found that nanoparticle scaffolds of HA with high affinity promoted the growth of additional capillaries and vascular networks [10]. Peiyang Gu et al. ectopic osteogenesis experiments in nude mice subcutaneously further demonstrated that HA promotes the growth of cells and biomineralization deposition within the thin filaments and significantly up-regulates the expression of genes and proteins associated with osteogenesis and angiogenesis-related genes and protein secretion [11]. Specifically, it was shown that HA/PCL composites encouraged bone marrow mesenchymal stem cells (BMSCs) to express bone morphogenetic protein 2 (BMP-2) and vascular growth factor [9].

In addition to the selection of appropriate filling materials for bone defect repair, the geometric spatial structure of the scaffold is equally important. Because bone cancellous matrix is a loose and porous three-dimensional structure, the structure of the scaffold pores forms the pathways for material transport and determines local nutrient supply, intravascular and host-graft interactions [12]. Therefore, it is crucial to combine the features of 3D printing to prepare spatially structured scaffolds with bionic bone. Furthermore, it has been shown that the immune microenvironment is inextricably linked to bone regeneration, and that scaffold-regulated polarization of macrophages (MPs) towards M2 promotes an immune microenvironment conducive to osteogenesis [13].

Hui Mei et al. [14] investigated the effect of different fibre angles on the mechanical properties of 3D materials and found that "0˚/45˚/90˚" had a higher tensile strength compared to "30˚/45˚/60˚" and "15˚/45˚/75˚". Barbara Ostrowska et al. [15]found that the internal geometry of the scaffold affected the tensile strength of human bone marrow mesenchymal stem cells (hMSCs). A study by Barbara Ostrowska et al15 found that the internal geometry of the scaffold affects the distribution of human bone marrow mesenchymal stem cells (hMSCs), and that the 60˚ fibre corner scaffold significantly promotes the expression of osteogenic genes by hMSCs compared to 90˚. However, there are fewer studies on 3D printed fibres towards simultaneous regulation of stem cells and macrophages. Therefore, the present study was devoted to the preparation of an efficient, inexpensive, and reproducible structure. We constructed a bionic 3D HA/PCL scaffold with multiple angles of fibre orientation and evaluated the effects of different fibre orientations on the physicochemical properties of the scaffold, osteogenesis and angiogenesis, and immunomodulation.

## Materials and methods

### Materials

PCL (Molecular weight: 54,000) was from Jinan Daigang Biotech Co. HA (Model: EFL-Nano-nHA-001, average particle size: 20 nm, purity: 99.9%) were purchased from Suzhou Institute of Intelligent Manufacturing (Suzhou, China). Ultrapure RNA Kit, Hi Fi Script c DNA Synthesis Kit, and Ultra SYBR Mixture (Low ROX) were purchased from CWBIO. Alkaline phosphatase (ALP) kit, BCA protein assay kit, and RIPA lysate were supplied by Biyun Tian Biotechnology Co. DMEM medium, fetal bovine serum, penicillin-streptomycin solution, and CCK8 kit were from Gibco. rat bone marrow mesenchymal stem cells (BMSCs) and macrophage cells (RAW264.7) were supplied by Gibco. Cells were supplied by ATCC (U.S.A.).

### Preparation of HA/PCL scaffolds

100 mg of HA powder was weighed and dispersed in 10 ml of dichloromethane solution containing 1 g of PCL. Pour into the appropriate glass dishes and evaporate the methylene chloride to obtain the HA/PCL scaffold material. The codes for the scaffolds with different fibre orientations (angles between the upper and lower layers of fibres) were written on a 3D printer (EFL-BP-6603, Suzhou Institute of Intelligent Manufacturing, Suzhou, China). We designed the scaffolds with a fiber size of 500 um and a fiber spacing of 900 um. The fiber pinch angles were 0˚, 45˚, 90˚, and 135˚, and the overall diameter of the scaffolds was 8 mm and the height was 1 mm. The raw materials were put into a 3D printer, heated and melted to print a porous cylindrical 3D scaffold. The obtained scaffolds were named as 0–90, 0-90-135 and 0-90-45.

### Surface morphology of the scaffolds

Three groups of scaffolds were attached to the operating table with conductive adhesive and the surface was sprayed with gold. The microscopic morphology and structure of the scaffolds were observed by scanning electron microscope (SEM) (Merlin) at an accelerating voltage of 10kV.

### Porosity and density testing of the scaffolds

The mass of the dried scaffolds sample was measured as $M_0$. The total mass of the specific gravity flask filled with anhydrous ethanol at room temperature was recorded as $M_1$, the total mass of the sample placed in the flask and ultrasonicated for 5 min to exhaust the air bubbles was measured as $M_2$, and the residual mass of the sample removed and weighed was recorded as $M_3$. The scaffolds obtained the porosity was calculated by the following formula:

$$P = (M_2 - M_3 - M_0)/(M_1 - M_3)$$

The diameter of 8 mm and height of 1 mm is the volume (V) of three sets of porous cylindrical scaffolds, and the density of the scaffolds is given by following formula:

$$\rho = M_0/V$$

### Mechanical strength test of the scaffolds

The compression modulus of the scaffolds was tested using a universal mechanical testing machine (Instron, 345C-S). At room temperature, the height of the scaffold was measured with a vernier caliper and the scaffold was placed on the test platform with the extrusion speed

set at 5 mm/min. The test was stopped when the scaffold had reached 80% of the compression deformation level. The stress-strain curves were plotted using Origin 9.0 software. The corresponding modulus of elasticity was obtained by calculating the slope of the linear part of the stress-strain curve.

## Hydrophilicity test of the scaffolds

Put the scaffolds to be measured on the lifting platform, adjust the height of the lifting platform, use the contact angle meter to drop the drop on the surface of the scaffolds, each drop amount is 2μl, adjust the position of the sample and the drop in contact with the baseline of the measurement, take a picture to obtain the static image of the drop at this time, use the contact angle measurement software to calculate the contact angle (θ) of the size of the scaffolds in each group.

## Protein adsorption

First, three groups of scaffolds were placed in 150 ug/ml bovine serum albumin (BSA/PBS) solution (500 uL) on an oscillator (25 rpm) for 6 h. After rinsing thoroughly with PBS, the scaffolds were homogenized with 1% sodium dodecyl sulfate solution and then centrifuged at 48°C for 15 min. Total protein in the supernatant was detected by BCA protein assay kit.

## Culturing and the scaffolds sterilization of BMSCs

Place the cryovial containing the BMSCs in a 37°C water bath and gently agitate to facilitate thawing. Using a sterile pipette, transfer the contents of the cryovial into a 15mL centrifuge tube, then centrifuge for 5min at 1000rpm. Discard the supernatant, resuspend the cells and transfer them to a 10cm2 dish and place them in an incubator at 37°C, 5% CO2 and saturated humidity, change the fluid regularly and observe the cell morphology. The cells were cultured at 80–90% fusion and passaged, and BMSCs within 4 generations were used for the experiments.

The scaffolds in each group were soaked in 75% ethanol solution for 2 h before the experiment, and the scaffolds were lubricated with PBS three times and overnight in a UV sterilizer (30L-800L, Shengzhiyuanhe Scientific and Educational Equipment Co., Ltd., Jiangsu China). A scaffold was placed in each well during the experiment.

## Cell proliferation and adhesion

Obtain BMSCs from the third generation in the logarithmic growth phase, and create a cell suspension by digesting with trypsin and adding growth medium. BMSCs are seeded onto the surfaces of the scaffolds at a density of $1 \times 10^5$ cells/cm$^2$ and cultured in a 48-well plate. After 1, 3 and 7 consecutive days of incubation, the medium was removed and the scaffolds were individually clamped into new 48-well plates with forceps and 10% CCK-8 solution was added. The plate is then incubated for 2 hours under conditions of 37°C and 5% $CO_2$. Finally, an enzyme labeling instrument (Tecan Spark, Austria) is used to measure the absorbance at 450nm to assess the proliferative capacity of the cells on each group of scaffolds.

Cells were inoculated and cultured as above. After three days of culture, the cells on the surface of the samples were stained with a live cell dye, and the morphology and distribution of the cells on the surface of the samples were observed by inverted fluorescence microscope and photographed for the determination of the viability of the bone marrow mesenchymal stem cells.

## Detection of MPs polarization gene expression levels

RAW264.7 cells were inoculated onto the scaffold samples of 48-well plates at a density of $5 \times 10^4$ cells per well. The complete culture medium was changed every day. On day 3, real-time fluorescence quantitative PCR was used to detect the expression of the M1 immuno-marker genes Tumor necrosis factor-alpha (TNF-α, Recombinant Human Interleukin-1 beta (IL-1β), and the M2 immunomarker genes CD206 and Arginase (Arg) expression. Glyceraldehyde phosphate dehydrogenase (GAPDH) was used as an internal reference gene.

## Evaluation of osteogenic and angiogenesis properties of the scaffolds

$1 \times 10^5$ BMSCs were added to 48-well plates containing the samples, Osteogenic induction solution was configured using DMEM complete medium at a ratio of 0.39 mg dexamethasone, 1.76 mg vitamin C, and 306.11 mg sodium β-glycerophosphate per 100 ml for culture. After 7 and 14 days of osteogenic induction culture, the induction was terminated, and real-time fluorescence quantitative PCR was used to detect the osteogenic marker genes Alkaline phosphatase (ALP), Runt-related Transcription Factor 2 (RUNX2), Collagen type I (COLl), Vascular endothelial growth factor (VEGF), Platelet-derived growth factor (PDGF), in the cells, and GAPDH were used as the internal reference gene. The sequences of the primers for the genes used are shown in Table 1.

The experimental results obtained were analysed using the $2^{-\Delta\Delta Ct}$ method. $-\triangle\triangle Ct$ was calculated as follows

$$\triangle\triangle Ct = (Ct_{\text{target gene}} - Ct_{\text{GAPDH}})^{\text{experimental group}} - (Ct_{\text{target gene}} - Ct_{\text{GAPDH}})^{\text{control group}}.$$

The relative expression (Fold change) of each gene in the experimental group relative to the control group is $2^{-\Delta\Delta Ct}$.

The scaffolds were stained with an ALP staining kit for ALP staining on the 7th day of osteogenic induction. The BCA method was used for ALP enzyme activity quantification. Briefly, cells in the well plates were lysed by adding cell lysate according to the Alkaline Phosphatase Assay Kit instructions, homogenized appropriately and subsequently centrifuged to extract the supernatant for ALP assay. The relevant working solution and samples were added sequentially in a 96-well plate, and the reaction was terminated by incubation at 37°C for 30 min, and the absorbance was measured at 405 nm.

**Table 1. Primers used in real-time quantification of selected gene transcripts.**

| Gene transcript | Forward primer sequence (5'-3') | Reverse primer sequence (5'-3') |
| --- | --- | --- |
| Glyceraldehyde phosphate dehydrogenase (GAPDH) | GCCATGAGGTCCACCACCCT | AAGGTCATCCCAGAGCTG |
| Alkaline phosphatase (ALP) | GGAGATGGTATGGGGCGTCTC | GGACCTGAGCGTTGGTGTTA |
| Runt-related Transcription Factor 2 (RUNX2) | TCGGAGAGGTACCAGATGGG | AGGTGAAACTCTTGCCTCGT |
| Osteopontin (OPN) | TGAAACTCGTGGCTCTGATG | GATGAACCAAGCGTGGAAAC |
| Collagen type I (COLl) | TTCTCCTGGCAAAGACGGAC | CTCAAGGTCACGGTCACGAA |
| Bone morphogenetic protein-2 (BMP-2) | ACCCGCTGTCTTCTAGTGTTG | TTCTTCGTGATGGAAGCTGAG |
| CD206 | ATGGATGTTGATGGCTACTGG | TTCTGACTCTGGACACTTGC |
| Arginase (Arg) | CATATCTGCCAAAGACATCG | GGTCTCTTCCATCACCTTGC |
| Tumor necrosis factor-α (TNF-α) | GGGTGTTCATCCATTCTC | GGTCACTGTCCCAGCAT |
| Recombinant Human Interleukin-1 beta (IL-1β) | TACAGGCTCCGAGATGAACA | AGGCCACAGGTATTTTGTCG |
| Vascular endothelial growth factor (VEGF) | CTGCCGTCCGATTGAGACC | CTCCCGTGGCTTCTAGTGC |
| Platelet-derived growth factor (PDGF) | CACAATAACGGGAGGCTGGT | CACCAGTTTGATGGGACGGGA |

## Statistical analysis

The experiments were repeated six times, and the results are expressed as the mean ± standard deviation. All statistical analyses were processed using GraphPad Prism software. One-way ANOVA followed by Dunnett's multiple comparison test was used to determine significant differences between the experimental groups. $P < 0.05$ was considered statistically significant.

## Results

### Test results of physical and chemical properties of the scaffolds

The surface morphology of the scaffold was observed by SEM, as shown in Fig 1E. The 0–90 has a fiber crossing angle of 90˚, a fiber size of about 500 um, and a fiber spacing of about 900 um. The 0-90-135 has an additional 135˚ fiber, while 0-90-45 is obtained by rotating the 0–90 by 45˚. From the high magnification microscope, the surface of the scaffold was all scattered with HA particles, and these particles increased the surface roughness, which facilitates cell adhesion. The difference in fibre orientation affected the physicochemical properties of the scaffolds. The porosity of 0–90, 0-90-135 and 0-90-45 were 82.687 ± 3.494%, 76.000 ± 1.113%, 72.665 ± 0.720% (Fig 1A), respectively, and the densities were 0.572 ± 0.001 g/cm$^3$, 0.615 ± 0.012 g/cm$^3$, 0.620 ± 0.006 g/cm$^3$ (Fig 1B). The difference in the angle of fibre entrapment increased the density of the scaffold while decreasing the porosity. In terms of compressive properties, the most significant was 19.640 ± 0.823 MPa for the 0–90 group and the lowest was 11.830 ± 0.480 MPa for the 0-90-135 group (Fig 1C), whereas there was no significant difference in the modulus of elasticity among the three groups (Fig 1D). The hydrophilicity of the scaffolds increased with the increase in the complexity of the fibre orientation (Fig 1F and 1G), with the smallest contact angle (72.825 ± 1.794˚) for 0-90-45, followed by 81.600 ± 0.528˚ for 0-90-135, and the largest for the 0–90 group (118.208 ± 1.412˚). And this is also consistent with the trend of protein adsorption properties of scaffolds Fig 1H), 0-90-45 had the most significant ability to adsorb BSA and the lowest was in 0–90 group.

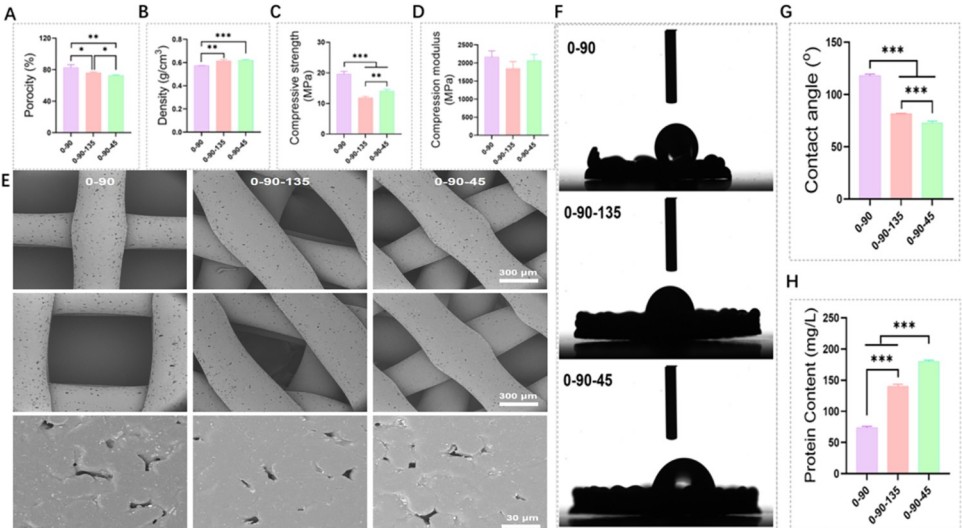

**Fig 1. Pictures of physical and chemical properties of 0–90, 0-90-135 and 0-90-45.** (A) Porosity test graph of the scaffolds. (B) Density test of scaffolds. (C and D) Compressive strength and modulus of elasticity of scaffolds. (E) SEM image of the scaffold. (F and G) Contact angle test results of the scaffold. (H) Protein adsorption properties of the scaffold.

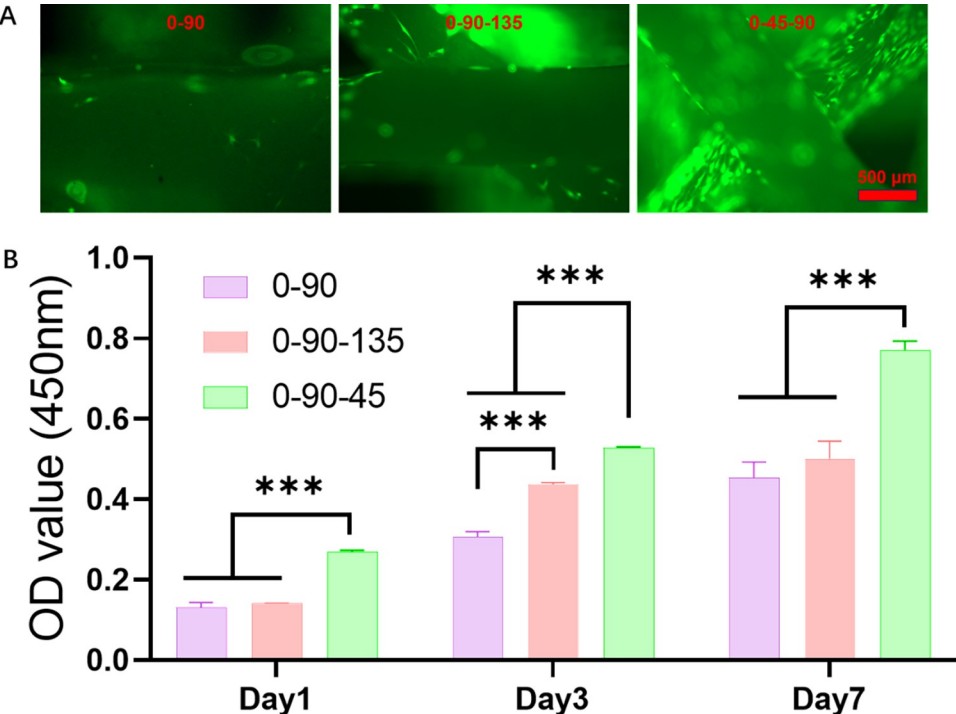

**Fig 2. Pictures of live cell staining and proliferation of BMSCs on scaffolds with different fibre orientations.** (A) Live cell staining results of BMSCs co-cultured with scaffolds for 3 days. (B) Proliferation activity graphs of BMSCs co-cultured with scaffolds for 1, 3 and 7 days.

## Different fibres towards scaffolds promote proliferation and adhesion of BMSCs

As shown in Fig 2B. On the 1st day of co-culture of BMSCs with scaffolds, the OD values of 0-90-45 group were significantly higher than the other groups, while the difference between the remaining two groups was not significant. On the third day, the proliferative activity of BMSCs showed a decreasing trend of 0-90-45, 0-09-135, and 0–90. This trend was consistent with the live cell staining on the third day (Fig 2A), where BMSCs significantly adhered to the 0-90-45 group, showing the strongest fluorescence. The cell proliferation of all scaffold groups increased with the increase of incubation time, and the 0-90-45 group remained significantly better than the other groups on day 7.

## Differences in the polarization properties of MPs promoted by different fibres towards the scaffolds

In terms of in vitro regulation of MPs in scaffolds, overall, the expression of pro-M1-type genes (TNF-α, IL-1β) was most significantly up-regulated in the 0–90 group, and the expression of pro-inflammatory genes was significantly down-regulated in the 0-90-45 group compared to the other groups (Fig 3A). In the expression of M2-type genes, de CD206 and ARG expression was significantly up-regulated in the 0-90-45 group, showing an increasing trend of 0–90, 0-90-135, 0-90-45 (Fig 3B).

## Differences in osteogenic differentiation and angiogenic gene expression properties of BMSCs promoted by different fibres towards the scaffolds

The expression results of 0–90, 0-90-135 and 0-90-45 pro-osteogenic differentiation-related genes (ALP, OPN, BMP-2, COL-1, RUNX2) are shown in Fig 4. During the process of

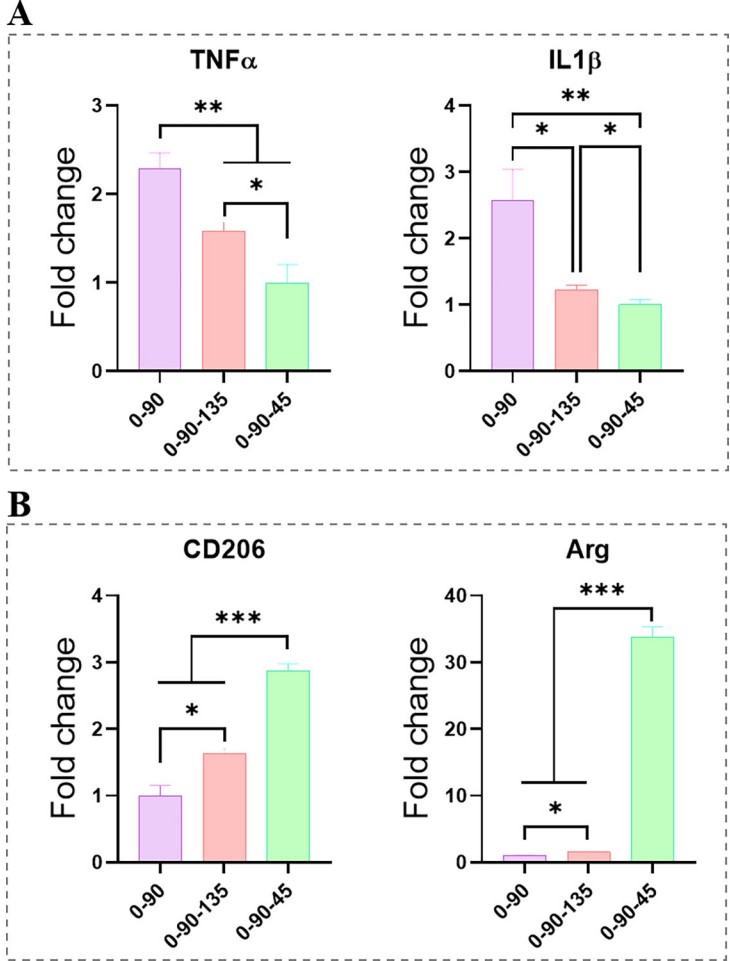

**Fig 3. Plot of differences in polarization properties of 0–90, 0-90-135 and 0-90-45 pro-MPs.** (A) Expression graph of scaffold-promoted MPs M1 phenotypic polarization genes (TNF-α, IL-1β). (B) Expression graph of scaffold-promoting MPs M2 phenotypic polarization genes (CD206, Arg).

osteogenic differentiation, the expression of 0-90-45-promoted related osteogenic genes was significantly higher than that of the other two groups, except that there was no significant difference in the expression of RUNX2 among the three groups of scaffolds at day 7. While the 0-90-135 group was generally superior to the 0–90 group. ALP quantification was performed on day 7 of osteogenesis induction (Fig 5B), and the highest ALP expression was found in the 0-90-45 group, followed by the 0-90-135 group, and the lowest was found in the 0–90 group, which was also consistent with the ALP staining results (Fig 5A). In angiogenic factors (PDGF, VEGF) expression demonstrated in Fig 6, the 0-90-45 group was significantly better than the remaining two groups at all time points. In PDGF expression, there was no significant difference between the 0–90 and 0-90-135 groups, whereas compared to the 0–90 group, the 0-90-135 significantly upregulated VEGF expression. Overall, 0–90 < 0-90-135 < 0-90-45 in terms of promoting osteogenic differentiation and angiogenic properties of BMSCs.

## Discussion

Scaffolds play the role of bridge connections in bone repair. In bone defects, the scaffold can mimic the bone extracellular matrix, providing support and conduction, while the internal

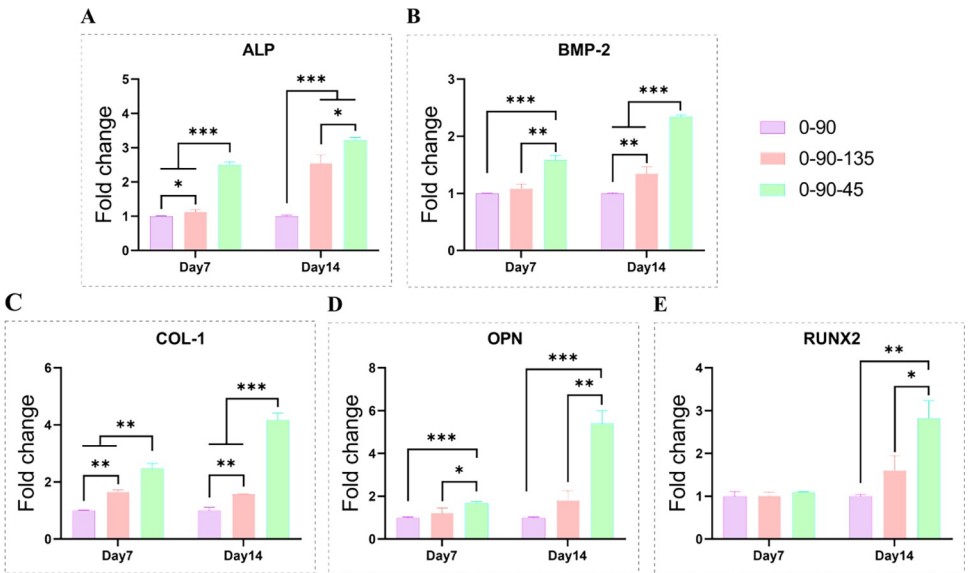

**Fig 4.** Expression maps of osteogenic genes (ALP, RUNX2, OPN, COLI, BMP-2) in 0–90, 0-90-135 and 0-90-45 co-cultured with BMSCs on days 7 and 14.

pores facilitate the implantation of new bone and blood vessels. 3D printing technology, a type of additive manufacturing, can be used to build predefined shapes of porous biocompatible scaffolds with good mechanical and osteoconductive properties using three-dimensional multilayers to accurately reconstruct the shape of bone defects [16]. In this study, scaffolds with different fibre orientations were prepared by 3D printing to mimic the structure of bone

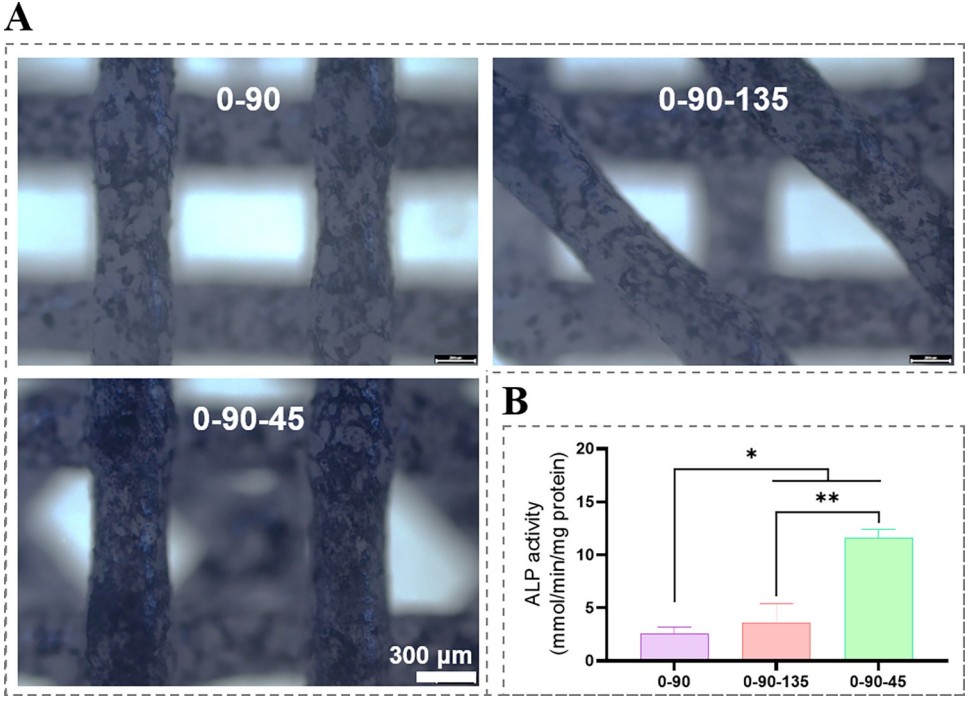

**Fig 5.** ALP staining (A) and quantitative analysis (B) of scaffolds co-cultured with BMSCs for 7 days.

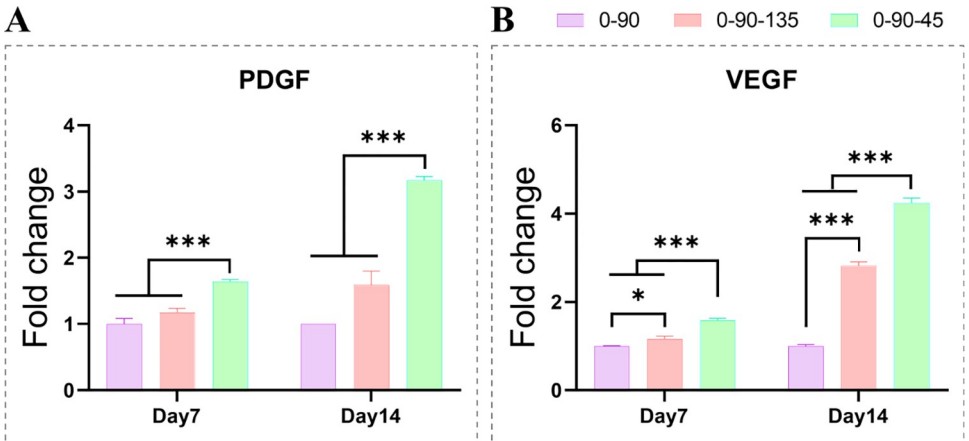

**Fig 6.** Expression of angiogenesis-related genes PDGF (A) and VEGF (B) in scaffolds co-cultured with BMSCs on days 7 and 14.

cancellous matrix, and we found that multicrossed fibres appeared to be more conducive to the induction of osteogenesis and angiogenesis.

Adequate porosity promotes the diffusion of nutrients and oxygen and the removal of wastes, ensuring that the scaffolding is bone-integrated [17,18]. In the three sets of scaffolds with the change of crossing angle made the original porosity changed, but all of them were greater than 70 per cent. It was noted that the porosity of the scaffolds could increase cell viability and promote bone growth compared to the porosity of human bone trabeculae (70%-90%), and therefore, high porosity (60%) is widely believed to promote osteogenesis [19]. In the compressive performance test, 0–90 has the most significant compressive strength, while 0-90-135 has the lowest (Fig 1C). This may be due to the fact that the 0–90 fibres have a uniform and homogeneous direction of force, whereas the 135° orientation of the 0-90-135 fibres alters the original force structure, and the compressive strength of the 0-90-45 is more significant because of its higher density than that of the 0-90-135 [20]. In addition, the more complex fibre orientation leads to an increased number of fibres per unit area, which is more conducive to protein and cell adhesion.

Contact of the bone filler material with body fluids initiates an immune response dominated by MPs, which is crucial for the success of the implant surgery [21]. The inflammatory microenvironment created by M1-type MPs promotes the progression of inflammation, whereas the activation of M2-type MPs activates more anti-inflammatory factors and promotes tissue repair [22]. Here, 0-90-45 better induced polarization of MPs to form an immune microenvironment rich in M2 phenotypes. We speculate that this may be due to the fact that the small angle of fibre entrapment allows the MPs to stretch sufficiently, and that the degree of MPs stretching correlates with the expression of M2 phenotype markers [23,24].

During bone regeneration, ALP activity is the most widely recognized marker of osteoblast activity, and BMP-2 is the most potent osteogenic factor known. Runx-2, a transcription factor at the downstream end of the bone morphogenetic protein (BMP) signaling pathway, is a major regulator of osteoblast development and bone formation, and is thought to be the first transcription factor to trigger osteogenic differentiation. COLI and OCN are involved in the regulation of calcium ion stabilization, induces BMSCs differentiation and favors bone mineralization [25,26]. Expression of angiogenic genes such as PDGF and VEGF can recruit endothelial cells to form vascularization at the defect and promote osteogenesis [27,28]. It has been better shown that the HA/PCL scaffold regulates the expression of osteogenic and angiogenic

genes, which may be related to the calcium and phosphorus ions released by HA [9,29]. However, 0-90-45 > 0-90-135 > 0–90 (Figs 4–6) in the performance of promoting the upregulation of the expression of these genes. Kenny Man et al. noted that high curvature of the scaffold surface or over sharpening of the fibre angle (small pinch angle) increases the tension and promotes the differentiation of BMSCs [30]. Sio-Mei Lien et al. found that pore sizes of 50–150 μm were suitable for cell proliferation, while cells with the largest pore sizes (250–500 μm) were more inclined to secrete extracellular matrix [29]. In our another study, compared to 800 μm, scaffolds with a pore size of 500 μm significantly promoted BMSCs differentiation [31]. Compared to the other groups, the 0-90-45 group showed higher surface curvature, smaller fiber pinch angle and relatively suitable pore size, and thus exhibited significant pro-BMSCs differentiation properties. This may also be due to the fact that the multi-angle interlaced fibres better mimic the physiological structure of cancellous bone, and the scaffolds are in an optimal balance between physicochemical properties, cell adhesion and nutrient exchange [32].

In summary, different fibres with bionic structure towards the scaffold can better regulate the polarization of MPs, and the synergistic raw material (HA/PCL) embodies a superior osteogenic potential. However, in vivo experimental validation of this study is still needed in the next step.

## Conclusion

In this paper, we prepared HA/PCL scaffolds with different fibre crossings (orientations) (0–90, 0-90-135, 0-90-45) aiming to simulate the three-dimensional structure of bone cancellous mass. The results showed that the multi-angle interlaced fibres affected the overall porosity and compressive strength of the scaffolds. Compared to the other two groups, the 0-90-45 scaffolds were able to induce the polarization of MPs towards the M2 phenotype to form an osteogenic-friendly immune microenvironment, and also possessed the most significant osteoblastic-angiogenic potential. In conclusion, bionicity is the life of the scaffold, and this study is a further step forward in the exploration of BET scaffolds.

## Supporting information

**S1 File.**
(ZIP)

## Author Contributions

**Conceptualization:** Jiaxiang Song, Shuai Huang, Xitao Linghu, Weikang Xu, Qingde Wa.

**Data curation:** Jiaxiang Song, Shuai Huang, Xitao Linghu, Weikang Xu, Qingde Wa.

**Formal analysis:** Jiaxiang Song, Shuai Huang, Xitao Linghu, Weikang Xu, Qingde Wa.

**Funding acquisition:** Weikang Xu, Qingde Wa.

**Investigation:** Jiaxiang Song, Shuai Huang, Xitao Linghu, Weikang Xu, Qingde Wa.

**Methodology:** Jiaxiang Song, Shuai Huang, Xitao Linghu, Weikang Xu, Qingde Wa.

**Project administration:** Hengpeng Wu, Zhenyu Wen, Xiang Li, Qiping Huang, Weikang Xu, Qingde Wa.

**Supervision:** Weikang Xu, Qingde Wa.

**Writing – original draft:** Jiaxiang Song, Shuai Huang, Xitao Linghu.

**Writing – review & editing:** Weikang Xu, Qingde Wa.

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
