## [Decision Letter · Decision Letter 0]

10 Oct 2024

PONE-D-24-360293D printing of different fibres towards HA/PCL scaffolding induces macrophage polarization and promotes osteogenic differentiation of BMSCsPLOS ONE

Dear Dr. Wa,

Thank you for submitting your manuscript to PLOS ONE. After careful consideration, we feel that it has merit but does not fully meet PLOS ONE’s publication criteria as it currently stands. Therefore, we invite you to submit a revised version of the manuscript that addresses the points raised during the review process.

The manuscript, while presenting many qualities, needs a serious work to strengthen some points in the text.

The authors need to pay attention to the remarks from both reviewers when modifying the manuscript, especially regarding the characterization of the material itself. Some criteria need to be considered and included in this characterization. Some specific data regarding the HA needs to be specified.

Some minor mistakes found through the text should be corrected

We look forward to receiving your revised manuscript.

Kind regards,

Christophe Egles, Ph.D.

Academic Editor

PLOS ONE

Journal Requirements: When submitting your revision, we need you to address these additional requirements. 1. Please ensure that your manuscript meets PLOS ONE's style requirements, including those for file naming. The PLOS ONE style templates can be found at https://journals.plos.org/plosone/s/file?id=wjVg/PLOSOne_formatting_sample_main_body.pdf and https://journals.plos.org/plosone/s/file?id=ba62/PLOSOne_formatting_sample_title_authors_affiliations.pdf 2. We note that the grant information you provided in the ‘Funding Information’ and ‘Financial Disclosure’ sections do not match.  When you resubmit, please ensure that you provide the correct grant numbers for the awards you received for your study in the ‘Funding Information’ section. 3. Thank you for stating the following financial disclosure: "This research was supported by the Key Program for Science and Technology Project of Guizhou Province (ZK [2021] 007), the Program for Science and Technology Project of Guizhou Province, Qiankehe Platform Talents ([2021] 5613), the Guangdong Province Science and Technology Plan Project (2024A1515012265 and 2022A1515140193)." Please state what role the funders took in the study.  If the funders had no role, please state: ""The funders had no role in study design, data collection and analysis, decision to publish, or preparation of the manuscript."" If this statement is not correct you must amend it as needed. Please include this amended Role of Funder statement in your cover letter; we will change the online submission form on your behalf. 4. Thank you for stating the following in the Acknowledgments Section of your manuscript: "This research was supported by the Key Program for Science and Technology Project of Guizhou Province (ZK [2021] 007), the Program for Science and Technology Project of Guizhou Province, Qiankehe Platform Talents ([2021] 5613), the Guangdong Province Science and Technology Plan Project (2024A1515012265 and 2022A1515140193)." We note that you have provided funding information that is not currently declared in your Funding Statement. However, funding information should not appear in the Acknowledgments section or other areas of your manuscript. We will only publish funding information present in the Funding Statement section of the online submission form. Please remove any funding-related text from the manuscript and let us know how you would like to update your Funding Statement. Currently, your Funding Statement reads as follows: "The author(s) received no specific funding for this work" Please include your amended statements within your cover letter; we will change the online submission form on your behalf. 5. We note that your Data Availability Statement is currently as follows: All relevant data are within the manuscript and its Supporting Information files. Please confirm at this time whether or not your submission contains all raw data required to replicate the results of your study. Authors must share the “minimal data set” for their submission. PLOS defines the minimal data set to consist of the data required to replicate all study findings reported in the article, as well as related metadata and methods (https://journals.plos.org/plosone/s/data-availability#loc-minimal-data-set-definition). For example, authors should submit the following data: - The values behind the means, standard deviations and other measures reported;- The values used to build graphs;- The points extracted from images for analysis. Authors do not need to submit their entire data set if only a portion of the data was used in the reported study. If your submission does not contain these data, please either upload them as Supporting Information files or deposit them to a stable, public repository and provide us with the relevant URLs, DOIs, or accession numbers. For a list of recommended repositories, please see https://journals.plos.org/plosone/s/recommended-repositories. If there are ethical or legal restrictions on sharing a de-identified data set, please explain them in detail (e.g., data contain potentially sensitive information, data are owned by a third-party organization, etc.) and who has imposed them (e.g., an ethics committee). Please also provide contact information for a data access committee, ethics committee, or other institutional body to which data requests may be sent. If data are owned by a third party, please indicate how others may request data access. 6. PLOS requires an ORCID iD for the corresponding author in Editorial Manager on papers submitted after December 6th, 2016. Please ensure that you have an ORCID iD and that it is validated in Editorial Manager. To do this, go to ‘Update my Information’ (in the upper left-hand corner of the main menu), and click on the Fetch/Validate link next to the ORCID field. This will take you to the ORCID site and allow you to create a new iD or authenticate a pre-existing iD in Editorial Manager. 7. Please amend either the abstract on the online submission form (via Edit Submission) or the abstract in the manuscript so that they are identical. 8. We note you have included a table to which you do not refer in the text of your manuscript. Please ensure that you refer to Table 1 in your text; if accepted, production will need this reference to link the reader to the Table.

Reviewers' comments:

Reviewer's Responses to Questions

**Comments to the Author**

1. Is the manuscript technically sound, and do the data support the conclusions?

Reviewer #1: Yes

Reviewer #2: Partly

2. Has the statistical analysis been performed appropriately and rigorously? 

Reviewer #1: Yes

Reviewer #2: Yes

3. Have the authors made all data underlying the findings in their manuscript fully available?

Reviewer #1: Yes

Reviewer #2: Yes

4. Is the manuscript presented in an intelligible fashion and written in standard English?

Reviewer #1: Yes

Reviewer #2: Yes

5. Review Comments to the Author

Reviewer #1: The manuscript is well-written and clearly communicates the study's objectives and findings. It provides a comprehensive overview of the research on 3D-printed HA/PCL scaffolds, highlighting key aspects such as fibre orientation, osteogenic induction, and macrophage polarization. The novelty lies in exploring different fibre orientations and their effects on scaffold properties and biological responses, which offers valuable insights into the field. This work could be considered for publication after addressing the following comments:

- Section 2.1: Please provide more information about the HA powder used (e.g., particle size, purity, etc.).

- Since fibre orientation could significantly affect the mechanical and physical properties of the fabricated scaffolds via the 3D printing method, please present the results shown in Fig. 1 in at least two or three separate figures for better illustration.

- Section 2.2: HA powder does not dissolve in dichloromethane; it is dispersed. Please revise accordingly.

- Section 2.2: What does “mm I.D. printing needle” mean?

- Gene expression is typically evaluated over one month, but the authors studied it for only 14 days. What is the reason for this shorter duration?

Reviewer #2: Dear Authors,

Dear EiC,

I am a bit confused after reading the paper. The biological part seems good but the characterization of the material is very ambiguous. In this form, I cannot recommend the publication. In my opinion, the 3D printed scaffold have to be better characterized and after this more correlations to be done with the biological data. Some of the most important idea I noticed during the lecture of the manuscript are listed bellow

HA was dispersed and not dissolved in DCM

It is strange at all that the angle between the strands is considered but the distance between these are not considered. Size is critical in bone tissue engineering and cells need pores of 50 -150 um to be able to get inside them and to assure a proper ingrowth. Even if some references are done related to pore size, this part is not at all described. Also, the porosity is greater than 70% but I am not sure if it is determined experimentally of just estimated based on the printing model.

How did you monitor the Ca and P release ?

How can we interpret this “Furthermore, in scaffolds with smaller pore sizes, the preference is for cell growth rather than extracellular matrix secretion” – smaller than which value? There are many sentences which are unclear and not at all defined well – not clearly quantifiable.

Minor English polishing is still needed. For instance, instead of bio ceramics it should be bioceramics; instead of “to free more calcium and phosphorus components to help bone mineralisation” it is better to use “to release more calcium and phosphorus components to help bone mineralisation”; why you added “/” in “/A study by Barbara Ostrowska et al15” …

Best regards,

R1

6. PLOS authors have the option to publish the peer review history of their article (what does this mean?). If published, this will include your full peer review and any attached files.

Reviewer #1: **Yes: **Ghasem Dini

Reviewer #2: No

---

## [Author Response · Author response to Decision Letter 0]

28 Oct 2024

Reviewer #1: The manuscript is well-written and clearly communicates the study's objectives and findings. It provides a comprehensive overview of the research on 3D-printed HA/PCL scaffolds, highlighting key aspects such as fibre orientation, osteogenic induction, and macrophage polarization. The novelty lies in exploring different fibre orientations and their effects on scaffold properties and biological responses, which offers valuable insights into the field. This work could be considered for publication after addressing the following comments:

- Section 2.1: Please provide more information about the HA powder used (e.g., particle size, purity, etc.).

Dear reviewer, we have added this section.

Since fibre orientation could significantly affect the mechanical and physical properties of the fabricated scaffolds via the 3D printing method, please present the results shown in Fig. 1 in at least two or three separate figures for better illustration.

Dear reviewer, we have added accordingly.

Section 2.2: HA powder does not dissolve in dichloromethane; it is dispersed. Please revise accordingly.

Dear reviewer, we have revised it.

Section 2.2: What does “mm I.D. printing needle” mean?

Dear reviewer, this was a writing error on our part and we have revised this part of the description.

Gene expression is typically evaluated over one month, but the authors studied it for only 14 days. What is the reason for this shorter duration?

Dear reviewer, the early regulation of BMSCs by scaffolding is more significant, and the significance of the difference between groups is weakened if the time is too long to account for. The current mainstream studies on osteogenic differentiation of BMSCs are also at the time points of day 7 and 14[1, 2].

[1] Y. Miao, X. Liu, J. Luo, Q. Yang, Y. Chen, Y. Wang, Double-Network DNA Macroporous Hydrogel Enables Aptamer-Directed Cell Recruitment to Accelerate Bone Healing, Advanced Science 11(1) (2024).

[2] X. Ren, Q. Zhou, M. Oberoi, R. Caprini, N. Dahan, B. Harley, J. Lee, Physisorption of Recombinant Osteoprotegerin on Nanoparticulate Mineralized Collagen Scaffolds Reduces Osteoclast Resorption Activity without Affecting Osteogenesis, Journal of Bone and Mineral Research 37 (2022) 24-24.

Reviewer #2: I am a bit confused after reading the paper. The biological part seems good but the characterization of the material is very ambiguous. In this form, I cannot recommend the publication. In my opinion, the 3D printed scaffold have to be better characterized and after this more correlations to be done with the biological data. Some of the most important idea I noticed during the lecture of the manuscript are listed bellow

HA was dispersed and not dissolved in DCM

Dear reviewer, thank you for the reminder, we have fixed it.

It is strange at all that the angle between the strands is considered but the distance between these are not considered. Size is critical in bone tissue engineering and cells need pores of 50 -150 um to be able to get inside them and to assure a proper ingrowth. Even if some references are done related to pore size, this part is not at all described. Also, the porosity is greater than 70% but I am not sure if it is determined experimentally of just estimated based on the printing model.

Dear reviewer, it is us who overlooked this part. When we wrote the programming file, we designed the fiber size of 500 um, and the spacing of the fibers is 900 um. 0-90 is the fibers intersecting at 90o, and the four fibers constitute an aperture with an area of 8.1×105 um2. 0-90-135 is the addition of a 135o fiber to 0-90, which divides the original aperture into two equilateral right triangles with a base side of 650 um, and the theoretical aperture area of 0-90-135 is 4.225×105 um2. The theoretical aperture area of 0-90-135 is 4.225×105 um2, while 0-90-45 can be regarded as the result of rotating 45o on the basis of 0-90, and the original aperture is divided into four triangles with a base of 400 um and a height of 356.25 um, with a theoretical aperture area of 2.85×105 um2. Specific fiber sizes, spacings, and aperture diameters may deviate from the design parameters at the time of printing, but the scaffolding parameters, but the scaffolds were all made in the same batch, which did not affect the trend of scaffold structure change and experimental results. In addition, the porosity of the scaffolds is more than 70%, which we obtained through the porosity test experiment.

How did you monitor the Ca and P release ?

Dear reviewer, the release of Ca and P ions was not monitored in this study, and we only extrapolated the reasons for HA regulation of BMSCs and HUVES behavior based on existing studies such as Yifan Niu, Carvalho, MS et al[3, 4].

[3] Y. Niu, L. Chen, T. Wu, Recent Advances in Bioengineering Bone Revascularization Based on Composite Materials Comprising Hydroxyapatite, International Journal of Molecular Sciences 24(15) (2023).

[4] M.S. Carvalho, J.M.S. Cabral, C.L. da Silva, D. Vashishth, Bone Matrix Non-Collagenous Proteins in Tissue Engineering: Creating New Bone by Mimicking the Extracellular Matrix, Polymers 13(7) (2021).

How can we interpret this “Furthermore, in scaffolds with smaller pore sizes, the preference is for cell growth rather than extracellular matrix secretion” – smaller than which value? There are many sentences which are unclear and not at all defined well – not clearly quantifiable.

Dear reviewer, our description is not rigorous, based on the results of other people's studies arbitrarily came to this conclusion, has been referenced to supplement, thank you for your correction. Sio-Mei Lien et al. found that pore sizes of 50-150 μm were suitable for cell proliferation, while cells with the largest pore sizes (250-500 μm) were more inclined to secrete extracellular matrix[4]. In our another study, compared to 800um, scaffolds with a pore size of 500um significantly promoted BMSCs differentiation[5].

[4] M.S. Carvalho, J.M.S. Cabral, C.L. da Silva, D. Vashishth, Bone Matrix Non-Collagenous Proteins in Tissue Engineering: Creating New Bone by Mimicking the Extracellular Matrix, Polymers 13(7) (2021).

[5] W. Huang, S. Huang, X. Linghu, W.-C. Chen, Y. Wang, J. Li, H. Yin, H. Zhang, W. Xu, Q. Wa, 3D-Printed mesoporous bioglass/ polycaprolactone scaffolds induce macrophage polarization toward M2 phenotype and immunomodulates osteogenic differentiation of BMSCs, 10(5) (2024).

Minor English polishing is still needed. For instance, instead of bio ceramics it should be bioceramics; instead of “to free more calcium and phosphorus components to help bone mineralisation” it is better to use “to release more calcium and phosphorus components to help bone mineralisation”; why you added “/” in “/A study by Barbara Ostrowska et al15” …Dear reviewer, we have revised it.

---

## [Editor Report · Decision Letter 1]

6 Nov 2024

3D printing of different fibres towards HA/PCL scaffolding induces macrophage polarization and promotes osteogenic differentiation of BMSCs

PONE-D-24-36029R1

Dear Dr. Wa,

We’re pleased to inform you that your manuscript has been judged scientifically suitable for publication and will be formally accepted for publication once it meets all outstanding technical requirements.

Kind regards,

Christophe Egles, Ph.D.

Academic Editor

PLOS ONE
---

## [Editor Report · Acceptance letter]

12 Nov 2024

PONE-D-24-36029R1 

PLOS ONE

Dear Dr. Wa, 

I'm pleased to inform you that your manuscript has been deemed suitable for publication in PLOS ONE. Congratulations! Your manuscript is now being handed over to our production team.

Kind regards, 

on behalf of

Professor Christophe Egles 

Academic Editor

PLOS ONE